# Sleep, Nutrition, and Injury Risk in Adolescent Athletes: A Narrative Review

**DOI:** 10.3390/nu15245101

**Published:** 2023-12-13

**Authors:** Lorcán Mason, James Connolly, Lydia E. Devenney, Karl Lacey, Jim O’Donovan, Rónán Doherty

**Affiliations:** 1Sports Lab North West, Atlantic Technological University Donegal, Port Road, F92 FC93 Letterkenny, Irelandronan.doherty@atu.ie (R.D.); 2Department of Computing, Atlantic Technological University Donegal, Port Road, F92 FC93 Letterkenny, Ireland; 3Faculty of Arts & Social Sciences, The Open University, Walton Hall, Milton Keynes MK7 6AA, UK; 4DCU Glasnevin Campus, Dublin City University, Collins Avenue Extension, Dublin 9, D09 Y8VX Dublin, Ireland; 5Sport Ireland Institute, National Sport Campus, Abbotstown, Dublin 15, D15 Y52H Dublin, Ireland

**Keywords:** adolescent, athletes, sleep, nutrition, injury risk

## Abstract

This narrative review explores the impact of sleep and nutrition on injury risk in adolescent athletes. Sleep is viewed as essential to the recuperation process and is distinguished as an active participant in recovery through its involvement in growth, repair, regeneration, and immunity. Furthermore, the literature has shown that the sleep of athletes impacts elements of athletic performance including both physical and cognitive performance, recovery, injury risk, and mental well-being. For sleep to have a restorative effect on the body, it must meet an individual’s sleep needs whilst also lasting for an adequate duration and being of adequate quality, which is age-dependent. The literature has suggested that athletes have increased sleep needs compared to those of the general population and thus the standard recommendations may not be sufficient for athletic populations. Therefore, a more individualised approach accounting for overall sleep health may be more appropriate for addressing sleep needs in individuals including athletes. The literature has demonstrated that adolescent athletes achieve, on average, ~6.3 h of sleep, demonstrating a discrepancy between sleep recommendations (8–10 h) and actual sleep achieved. Sleep–wake cycles undergo development during adolescence whereby adaptation occurs in sleep regulation during this phase. These adaptations increase sleep pressure tolerance and are driven by the maturation of physiological, psychological, and cognitive functioning along with delays in circadian rhythmicity, thus creating an environment for inadequate sleep during adolescence. As such, the adolescent period is a phase of rapid growth and maturation that presents multiple challenges to both sleep and nutrition; consequently, this places a significant burden on an adolescent athletes’ ability to recover, thus increasing the likelihood of injury. Therefore, this article aims to provide a comprehensive review of the available literature on the importance of sleep and nutrition interactions in injury risk in adolescent athletes. Furthermore, it provides foundations for informing further investigations exploring the relation of sleep and nutrition interactions to recovery during adolescence.

## 1. Introduction

Sleep can be defined as a reversible behavioural state of impaired consciousness through a reduction in sensory and motor activity [1,2,3,4,5]. Sleep is regarded as an active regulatory process [6] and facilitates the proper functioning of the brain and cognitive performance while also regulating physiological functions including substrate and energy metabolism [7,8,9], cardiovascular function [10,11], appetite [12], endocrine function [13,14], and immune function [15]. Several facilitating theories for sleep have been hypothesised [5,16], including the regeneration of immune and endocrine function, the nervous system, and metabolic cost of living, and aids in cognitive development and synaptic plasticity [17]. As such, sleep is viewed as essential to the recuperation process [3,18] and is distinguished as an active participant in recovery through its involvement in growth, repair, regeneration, and immunity [5]. Furthermore, the literature has shown that the sleep of athletes impacts elements of athletic performance including both physical and cognitive performance, recovery, injury risk, and mental well-being [4,19,20,21,22,23].

The architecture of sleep is organised into multiple 90-min series of one rapid-eye-movement (REM) and three non-REM sleep phases (N1, N2, and N3) [24,25,26]. Non-REM and REM sleep are distinct in nature and are characterised by brainwave activity, eye movement, cardiac rhythm, muscle tone, breath rate, and arousal thresholds [27,28]. REM sleep facilitates neurological regeneration, learning, memory, and emotional regulation [27,29,30]. Non-REM sleep is a three-phased process whereby the propensity to wake (the arousal threshold) is lowest during the first phase and progressively increases through to the highest point in the final phase (N3) [27,28] and can be characterised by the wave activity of the brain [27,31]. Non-REM sleep functions to support the regeneration of the nervous system, conserve energy, release anabolic hormones that augment protein synthesis to facilitate muscle recovery [32,33], and mobilise free fatty acids for ATP production [34]. For a detailed breakdown of the brainwave characteristics of the sleep cycle, readers should refer to the AASM manual for the scoring of sleep and associated events [25], and the principles and practices of sleep medicine [27].

## 2. Importance of Sleep Health

Sleep health is defined as an individualised and context-specific multidimensional pattern of sleep and wakefulness that supports physical and mental well-being [35] and is an integral component of not only living a healthy lifestyle [36,37] but also of adaptation and recovery [3,27,32,38,39]. The relationship between sleep and recovery in athletes can be viewed in terms of three key factors that affect restoration processes: 1. sleep duration (total sleep requirements including napping); 2. sleep quality (total sleep absent of sleep disorders, environmental disturbances, or sleep fragmentation); 3. sleep phase (circadian timing of sleep during the light–dark cycle) [39,40]. During adolescence, the psychosocial and societal pressures experienced may result in adverse sleep health and reduced recovery capacity [41,42]. This is due to sleep deficiencies, which have a negative impact on health and are linked to increases in all-cause mortality and disease risk [43]. Sleep deprivation (an insufficient sleep duration compared to the basal level) and disturbances (the inability to initiate and/or maintain the sleep–cycle) are also risk factors for adverse health, recovery, and injury risk in athletic populations [39,40,44,45,46,47,48].

For sleep to have a restorative effect on the body, it must meet an individual’s sleep needs whilst also lasting for an adequate duration and being of adequate quality, which is age-dependent [46]. Sleep needs can be defined as the optimum quantity of sleep required to maintain alertness and function throughout daily living [5]. Sleep duration can be influenced by exogenous and endogenous environmental characteristics, which adds complexity to defining an “optimal” sleep pattern due to high inter-individual differences [49,50,51]. Peripheral tissues contain molecular clocks within each cell that dictate the expression of clock-controlled genes in a period (the required time for a cortical neuron oscillation) or phase (waking time in relation to the light cycle) [49,52]. Processes controlled by circadian rhythms also influence sleep duration; thus, the time at which sleep occurs in the light cycle also has an integral role in sleep duration [50,53,54]. The difference between sleep needs and actual sleep duration is known as sleep debt [5].

## 3. Sleep Adaptations during Adolescence

The chronotype that defines the expression of individual circadian rhythmicity [55] may similarly shift during adolescence [56,57,58,59,60] due to the greater robustness to increased sleep pressure [61] and environmental factors that increase evening alertness [42]. An individual’s chronotype is mainly dictated by their genetic makeup; however, environmental and societal factors also affect the chronotype [55,62]. Cross-sectional research has evidenced that during adolescence, the distribution of the chronotype shifts toward the evening chronotype, reverting back to the earlier chronotypes post-maturation [56,57,58,59,60]. As sleep needs change over the lifespan, The National Sleep Foundation has published guidelines for age-dependent sleep durations, which includes recommendations for the following: adolescents (8–10 h), adults (7–9 h), and older adults (7–8 h) [46]. The literature has suggested that athletes have increased sleep needs, and thus the general recommendations may not be sufficient for athletic populations [23,63]. Therefore, a more individualised approach accounting for overall sleep health may be more appropriate for addressing sleep needs in individuals including athletes [23,35,63]. To feel rested, the literature has demonstrated that elite athletes need ~8.3 h of sleep [64,65]. Moreover, adolescent athletes achieve, on average, ~6.3 h [47,66,67] of sleep, demonstrating a discrepancy between sleep recommendations and actual sleep achieved [42,68,69]. Sleep–wake cycles also undergo development during adolescence whereby adaptation occurs in sleep regulation during this phase [68]. These adaptations to sleep regulation increase sleep pressure tolerance (reduced adenosine accumulation) [70,71,72] and are driven by the maturation of physiological, psychological, and cognitive functioning along with delays in circadian rhythmicity [73], thus creating an environment for inadequate sleep influenced by external factors associated with adolescence (Figure 1) [41,42,73,74]. This results in ever-decreasing time spent asleep during the ages of 15–18 years, with research reporting a decrease of ~1.5–3 h during this period [70,75,76]. Despite this, an adolescents sleep needs (pressure dissipation) under free living conditions does not appear to alter from the recommended ~9.25 h [42], irrespective of maturation status [77,78,79], thus attributing the decline to environmental factors as opposed to biological factors [73,80].

## 4. Growth, Maturation, and Energy Demands in Adolescent Athletes

Adolescence is the transitional life stage where the process of maturation occurs [81]. Maturation signifies the progressive period toward the adult or mature state [82], and is characterised by status (maturity state at the time of observation), timing (biological age at which specific maturational events occur), and tempo (the rate of maturational progression) [82,83,84,85,86]. During the maturation period, approximately 20% of the final adult height is reached and 50% of the predicted adult body weight is achieved with an increase of up to 40% in bone mass [81,87]. The adolescence period is a significant life stage that begins in conjunction with the onset of puberty. Puberty is unique to the individual with a vast range of inter-individual differences in maturation status impacting both physical and psycho-social development [82,84,86]. As evidenced, maturation status influences the development of and improvement in locomotive competencies in both a linear (accrual of strength capabilities) and non-linear (accrual of co-ordinative capabilities) fashion [88], influencing training adaptations in adolescent athletes [88,89,90,91]. Puberty onset is the catalyst of the growth period where the body undergoes meaningful physical and psycho-social adaptations such as alterations to body composition, including the accrual of bone, muscle, and fat mass, metabolic and endocrine system development, the development and maturation of the organ system, the establishment of nutrient storage and partitioning, and the establishment of self-esteem and psychological well-being, all of which affect general health and well-being [81,92,93,94]. Maturation results from the outcome of a multitude of complex processes that are governed by genetics, the endocrine system, environmental constraints, and nutrient intake [86,89,95,96].

## 5. Differences between Adolescent and Adult Athletes

During adolescence, there are vast inter-individual differences in nutritional needs dependent on factors including maturation status, body composition, physical activity, chronological age, and gender [92,95,96,97,98]. The implementation of correct and individualised nutrition for adolescent athletes not only supports overall health, adaptation, recovery, and athletic performance, but is also necessary for meeting growth and development demands (Table 1) [92,95,96,97,98]. Compared to their adult counterparts, adolescents possess several differences in substrate storage and metabolism, in conjunction with numerous physiological and metabolic alterations associated with maturation that contribute to an individual’s nutrient requirements [96]. Furthermore, nutrient and energy requirements in adolescence are also largely dictated by the interplay between three main factors: (1) current anthropometry, (2) maturation state, rate, and timing, and (3) physical activity and sporting demands [92,96,97]. These energy requirements of an individual are fulfilled by the intake of energy-yielding macro-nutrients, carbohydrates, protein, and fat [96,99,100,101]. The specific energy intake of adolescent athletes should be largely dictated by total daily energy expenditure (TDEE) [96,102,103,104].

Growth and maturation are energy-intensive processes where energy intake influences both the synthesis of new tissue and the deposition of nutrients into the new tissue [92,96,119]. The basal metabolic rate denotes the energy expended to synthesize new tissue during growth; however, energy deposition is difficult to accurately measure [92,96,119,120]. During both peak height velocity (PHV) and peak weight velocity (PWV; weight denotes body mass) periods, energy requirements are in flux and are variable among adolescents of the same chronological age, in particular their basal metabolic rates, which rapidly increases in a stepwise fashion to match their maturation status, timing, and tempo [95,96,121,122]. The thermic effect of activity often makes the largest contribution to energy requirements in adolescent athletes [92,96,121]. The thermic effect of activity is influenced by anthropometry and the duration, intensity, and mode of physical activity, which also contribute to total activity energy expenditure [92,96], resulting in large interindividual variability in energy requirements [92,95,96], creating difficulties in prescribing energy requirement recommendations for adolescent athletes [92,95,96,97]. Persistent low energy availability (LEA) contributes to negative outcomes in growth and development including impaired cellular, organ, and tissue development, reduced bone mineral density, an increased risk of stress fractures, delay and/or regression in sexual maturation, and immune deficiencies [96,123,124]. Thus, it is recommended that if any signs or symptoms of LEA persist in adolescent athletes, energy intake should be increased to prevent detrimental effects on maturation, recovery, and injury risk [96,103,124].

## 6. Nutrition Knowledge of Adolescents

As established within the literature, adequate nutrition is paramount for performance, recovery, and adaptation to training, along with optimising the maturation process during adolescence [96,103,125]; thus, sufficient nutritional knowledge is required to optimise dietary behaviours to support these processes and inform eating habits [126,127,128,129,130,131]. In the absence of sufficient nutritional knowledge, nutritional intake may be compromised due to poor food selection and decreased dietary quality [132]. This may negatively impact the training–recovery cycle in athletes and adolescent growth and development [126,127,129,130,131,133,134,135,136]. It has been reported that athletic populations fail to meet the recommended nutritional requirements to support training demands [130,132]; however, with regards to the level of nutrition knowledge, a weak positive relationship (r = >0.26) exists between the level of one’s nutrition knowledge and their energy balance and dietary quality [127,132,135,136]. Due to the multiple assessment tools utilised in nutrition knowledge research including the Abridged-Nutrition for Sport Knowledge Questionnaire (A-NSKQ), General and Sports Nutrition Knowledge Questionnaire (GeSNK), Nutrition Knowledge Questionnaire for Athletes (NKQA), Nutrition for Sport Knowledge Questionnaire (NSKQ), Nutrition Knowledge for Young and Adult Athletes (NUKYA), and Platform to Evaluate Athlete Knowledge of Sports Nutrition Questionnaire (PEAKS-NQ) [130], research is difficult to infer. Despite this, research suggests that that the mean correct scores for general (GNK) and sport (SNK) nutrition knowledge in adult athletes are between 40.2 ± 12.4% and 70 ± 9% [132]. Moreover, research in adolescent athletes suggests that mean nutrition knowledge ranges between 43.8 ± 11.4% and 48.85 ± 12.7% [129,137,138], which is lower than that of their elder counterparts (overall NK% = 55.1 ± 10.7%) [132], therefore highlighting the importance of increasing the level of nutritional knowledge in adolescent athletes to support maturation, recovery, and injury risk.

## 7. Recovery, Adaptation, and Fatigue during the Training Process

The aim of the training process is to progressively develop the required qualities of the sport in order to improve performance [139]. This is achieved via the balance between the application of an appropriate training dose and the time afforded to facilitate adequate recovery for sustained adaptations [140,141]. In essence, for adaptations to occur, an overloading training dose must be applied to the individual and homeostasis must be disturbed, resulting in reduced performance and fatigue [142]. Fatigue as a concept is extremely difficult to define due to its multifaceted origins [139], with numerous definitions proposed in the literature [143,144]. Despite this ambiguity, there is mutual agreement that a central component of fatigue is the failure to produce or maintain the required force or power output for a given task that was previously attainable resulting from both central and peripheral factors [143], including the activation of the motor command, propagation of the action potential through the descending motor pathway, myofilament excitation–contraction coupling, and the status of the intracellular milieu [145], which can persist for days at a time if not addressed [146]. However, despite this agreement, there is a failure to acknowledge the mental component of fatigue, which must be considered [143,146] due to the suggestion that when fatigue is reported as a symptom by an individual, it can only be evaluated via self-reporting and categorised as a trait characteristic or state variable [146]. Therefore, fatigue can be defined as a state in which an individual experiences an impairment of physical performance, mental fatigue, or excessive psychological distress [147]. Thus, if appropriate recovery is afforded following an appropriate training dose, adaptations occur that are protective against further fatigue arising from a similar training dose [148,149,150]. However, to fully explain the training–recovery cycle, practitioners must also account for the multitude of additional internal psycho-physiological responses and adaptations that also occur during training that dilute the accuracy of the training–recovery cycle (Figure 2) [151,152,153], resulting in a complex relationship between the training dose, performance outcomes, injury, and illness [154,155]. Therefore, a multi-dimensional approach to evaluate the individual response to the implied stressor is an essential part of the training–recovery cycle [151,152,153]. Ultimately, it is the athletes internal environment that determines the level of stress that governs the individual response to the implied training stressor [152]. In view of that, when monitoring the training–recovery cycle, it is recommended that measures of internal load are used as the primary means of determining the training adaptations [152]. This is since the internal load borne by an individual athlete corresponding to a specific stimulus will vary depending on the specific contextual factors of their internal environment, such as achieving sufficient sleep and energy balance, along with the nature of the sport [152,156,157,158,159], thus highlighting the importance of both sleep and nutrition in facilitating recovery to minimise an individual’s risk of injury [32,45,102,160,161,162,163,164].

## 8. Injury Risk in Adolescent Athletes

Sustaining an injury during training or competition is an inherent risk for an athlete. These injuries are a financial burden for sporting organisations in elite sport [165,166] and more importantly interfere with an individual’s ability to train or achieve optimal performance during competition [139,167,168]. Specifically for adolescent athletes, injuries present a risk to athletic advancement, health, and the enjoyment of participation in sport [83,169,170]. Elite youth sport places an added burden on adolescents due to the associated high training volume, increased training intensity, and demanding competition schedules [171,172,173]. As such, increased exposure to elite sports is likely to increase musculoskeletal injury risk [83,170,173], which may be exacerbated during peak growth spurts [173,174,175]. Thus, injury prevalence in adolescent athletes has increased in recent years [176] with an injury occurrence rate of ~1.4–6.4 per 1000 h during training and ~22.4 per 1000 h during competition [171,177,178,179]. Thus, identifying modifiable risk factors which can support recovery and mitigate injury risk are of utmost importance [45,154,155,162,163,167,168,180,181,182]. Both sleep and nutrition are viewed as modifiable facilitators for recovery in athletes [22,38,45,65,162,163,164,168,183], thus emphasising their relevance to injury mitigation interventions. As evidence, the adolescent period is a phase of rapid growth and maturation that presents multiple challenges to both sleep and nutrition [42,68,90,95,103,125]; consequently, this places a significant burden on an adolescent athlete’s ability to recover, thus increasing the likelihood of injury [45,67,162,163,178,184,185,186,187].

## 9. Relationship between Sleep, Nutrition, and Injury Risk in Adolescent Athletes

Despite the establishment of sleeps associations with injury risk in athletic populations [23,39,65,164,168], limited research has been conducted regarding adolescent athletes [45,67,162,163,186,188]. Research using subjective questionnaires have purported that adolescent athletes who experience < 8 h of sleep per night are 1.7 times (95% CI; 1.0–3.0; *p* = 0.04) more likely to sustain an injury [45]. Furthermore, research has found that decreasing hours of sleep during periods of high-volume intense training resulted in a 2.25-fold (95% CI; 1.46–3.45; *p* < 0.001) increase in the likelihood of sustaining an injury [163]. Moreover, solely accounting for a decreased sleep volume resulted in a 1.46-fold (95% CI; 1.10–1.94; *p* < 0.01) increased risk, while adolescent athletes who specifically reported that obtaining < 8 h of sleep resulted in a 1.31-fold (95% CI; 0.97–1.78; *p* = 0.080) increased injury risk [163]. More recently, research conducted in adolescent track and field athletes (12–21 years) aimed to investigate sleep as a predictor of injury using actigraphy [186]. Wake after sleep onset (WASO), which represents sleep disruption, was found to be a predictor of previous injury (OR = 1.144), while time spent awake (TA) was found to predict injury occurrence (OR = 0.974) in this cohort [186]. Furthermore, the researchers found that athletes who increased TA by at least 1 min reduced their likelihood of sustaining future injury ([F(2.36) = 6.512; *p* = 0.004]) [186]. Notwithstanding the importance of appropriate nutrition during adolescence [95,103,122,187,189] and nutrition’s influential interaction with sleep and recovery [38,164,183], limited research is available investigating the relationship between sleep, nutrition, and injury risk in adolescent athletes. Despite this, the interaction between sleep and nutrition cannot be understated with specific nutritional interventions, including a high-carbohydrate, high-glycaemic-index evening meal, melatonin supplementation, tart cherry juice, kiwifruit, and foods rich in tryptophan, all supporting proper sleep [38,164,183], which may have a positive impact on recovery and subsequently injury risk. However, in research investigating this interaction in adolescent athletes, associations have been found between diet quality, sleep, and injury risk [162]. Using subjective questionnaires in a population of 340 elite adolescent Swedish athletes, researchers found that athletes who reached the recommended nutrition intake decreased their injury risk by 64% (OR, 0.36; 95% CI, 0.14–0.91) [162]. Furthermore, it was reported that athletes who slept for more than 8 h per weeknight decreased their injury risk by 61% (OR, 0.39; 95% CI, 0.16–0.99) [162]. Moreover, during a competitive season, the duration, intensity, and frequency of training are strategically periodised as part of the training cycle [95,152,161,187]. This cyclical cycle results in periods of high and low training demand [161,190,191,192,193], which can impact elements of recovery including sleep and nutrition to facilitate adaptations to the training stimulus [139,151,152,153]. As demonstrated, adolescent athletes who achieve >8 h of sleep per night have a reduced injury risk [45,163,186]; thus, the literature suggests that during periods of high training demands, such as pre-season, adolescent athletes should achieve the recommended minimum of 8–10 h per night to facilitate proper recovery [23,163,194]. Furthermore, it is recommended that nutrition is prescribed in a periodised fashion to match these periods of high training demands and/or increased maturation to provide sufficient energy intake to support recovery [96,122,161,189].

## 10. Limitations

To date, limited research in the literature is available investigating the impact of both sleep and nutrition on injury risk in adolescent athletes. As such, the consensus of the available literature is mainly informed by studies involving adult athletes and/or general adolescent populations. Moreover, there is a scarcity of available literature on objective sleep measures such as actigraphy or polysomnography in adolescent athletes to inform current knowledge, and therefore the current conclusions are inferred from subjective questionnaires and sleep diaries. Furthermore, much of the available literature involves limited sample sizes, case studies, or cross-sectional investigations, which limits the ability to draw conclusions, and therefore future research investigating objective sleep measures longitudinally is warranted.

## 11. Conclusions

As has been established, both sleep and nutrition play an important role in recovery and injury risk in athletic populations [23,38,126,162,163,164,183]. As maturation is an energy-intensive process [92,96,119] coupled with the high-energy and training demands of adolescent athletes [92,95,96,97] and increased injury risk during periods of peak growth and/or training intensity [90,192,195,196,197], the recovery process cannot be understated. Moreover, with the apparent sleep adaptations including increased sleep pressure tolerance and circadian phase delay that occur during adolescence, together with the current research showing that compromised sleep (OR, 0.39; 95% CI, 0.16–0.99) and an inadequate diet (OR, 0.36; 95% CI, 0.14–0.91) may increase the likelihood of injury occurrence [45,67,162,163,186,188], further investigations are required to investigate the relationship between sleep, nutrition, and injury risk before sound conclusions can be made. Despite this, the literature does demonstrate a clear negative impact of poor sleep on injury risk in adolescent athletes [45,67,162,163,186,188]; similarly, research does appear to support the positive role of adequate nutrition on both sleep and injury risk [162]. Further research is warranted that investigates the impact of both objective sleep measures and specific nutritional interventions on injury risk in adolescent athletes.

## Figures and Tables

**Figure 1 nutrients-15-05101-f001:**
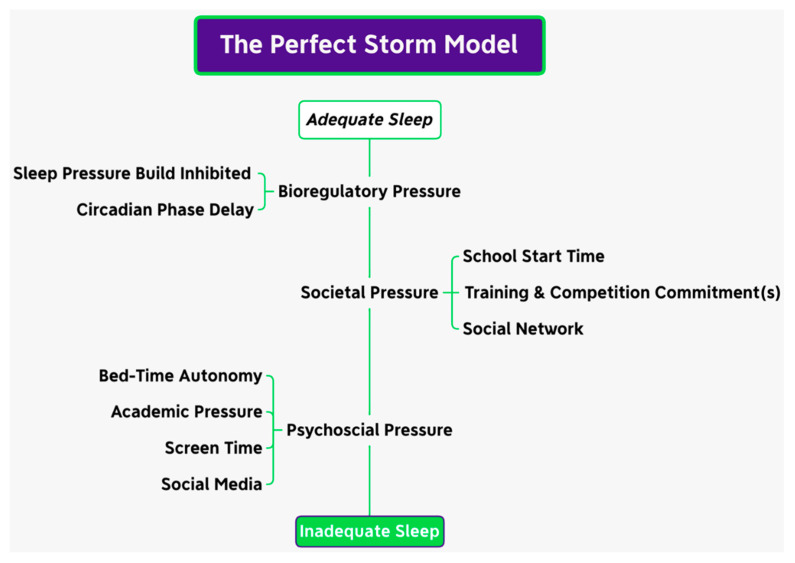
The perfect storm model adapted from [41,42].

**Figure 2 nutrients-15-05101-f002:**
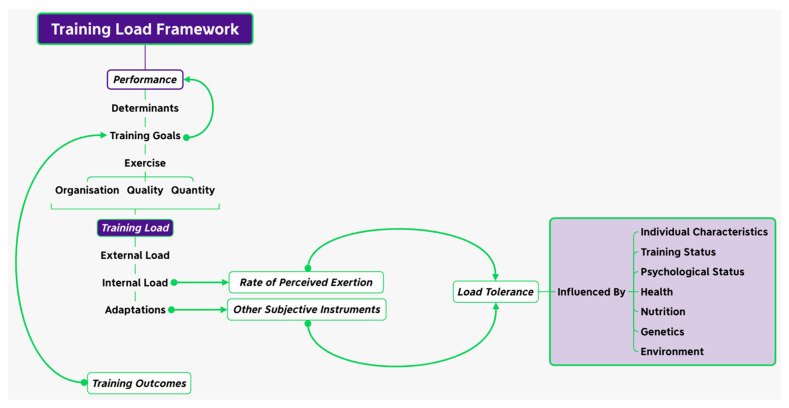
Training load framework adapted from [152].

**Table 1 nutrients-15-05101-t001:** Anatomical, physiological, and metabolic differences between adolescent and adult athletes adapted from [96].

Summary of Main Physiological and Metabolic Issues Surrounding Growth and Maturation	Potential Consequences of These Differences in Physiology and Metabolism on Nutritional Recommendations
Greater Energy Cost of Movement	
Children and adolescents have a higher (relative) energy cost of movement compared with that of adults. This may be due to increased stride frequency, a greater surface area:volume ratio, a more distal distribution of mass in the legs, or greater levels of contraction of the antagonist leg muscles while moving [105,106].	Increased (relative) energy requirements for physical activity need to be accounted for.
Reduced Glycogen Storage Capacity	
Children and adolescents have a lower endogenous glycogen storage capacity compared with that of adults [107].	Reduced emphasis for young athletes to have a carbohydrate load before training/competition.
Reduced Glycolytic Capabilities	
Children and adolescents have reduced glycolytic capabilities, with full anaerobic capabilities developing towards the end of puberty [108]. As a result, children and adolescents have lower levels of lactate production than those of adults during high-intensity exercise of the same relative intensity [107,109].	Reduced requirement for the use of buffering agents with young athletes, particularly those in pre- and peri-puberty stages.
Higher Rates of Aerobic Metabolism	
Higher rates of aerobic metabolism exist in children during exercise. Fat oxidation rates during submaximal exercise (of the same relative intensity) are greater in children and adolescents compared with that in adults. Less mature children have a greater reliance on fat as a fuel compared with more mature adolescents. It has been suggested that these higher fat oxidation rates in children compared with those in adults are the result of lower endogenous carbohydrate stores and reduced glycolytic capabilities [110].	Young athletes may not require the same relative amount of carbohydrate as adult athletes do; however, there is a lack of evidence to support this. Further research is warranted.
Greater Reliance on Exogenous Carbohydrate	
Children and adolescents have greater reliance on exogenous carbohydrate as a fuel source. During exercise, exogenous carbohydrate is a greater contributor to total energy supply in children and adolescents compared with adults [110]. Exogenous carbohydrate oxidation rates are higher in less mature boys compared with more mature boys of the same chronological age; however, this is not the case in females [111,112].	Exogenous carbohydrate should be consumed during moderate-/high intensity exercise lasting longer than −60 min.
Thermoregulatory Differences	
Children and adolescents have a larger surface area:body mass ratio [113], so, consequently, they gain and lose more heat from the environment through conduction, convection, and radiation. Adolescents who undertake regular exercise do adapt, however, improving their ability to thermoregulate through enhanced peripheral vasodilatation [114].	Regular consumption of cold flavoured fluids during exercise
Reduced Sweating Capacity	
Children and adolescents have a lower sweating capacity compared with that of adults and therefore a reduced ability to lose sweat through sweat evaporation. As children mature, so too do their thermoregulation mechanisms (particularly their ability to sweat); however, these are not fully developed until late puberty [115].	Regular consumption of cold flavoured fluids during exercise. There is no evidence to suggest that fluid requirements in young athletes are less than those of their adult counterparts, despite reduced sweat rates.
Growth and Increase in Body Size	
Macronutrient requirements are often prescribed relative to body mass (i.e., grams per kilo, g/kg) to account for individual differences in size among young athletes. Although fat mass does not seem to significantly change throughout growth and maturation in young athletes, increases in body mass are primarily derived from an increase in fat-free mass [116]. An increase in stature is the result of skeletal growth and the laying down of bone mineral content (i.e., skeletal tissue). Around 95% of adult bone mineral content is achieved by the end of adolescence, with ~26% of this being accrued at a peak bone mineral content velocity (~12.5 and ~14 years old in girls and boys respectively) [117]. Changes in fat-free mass and stature are significantly influenced by the energy and macronutrient intake of a young athlete during childhood and adolescence [118].	Increased (relative) energy requirements need to be accounted for during peak weight and height velocity periods.

## Data Availability

Data will be made available upon request.

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
