# Peer review of "Sleep, Nutrition, and Injury Risk in Adolescent Athletes: A Narrative Review"

_nutrients, 2023, doi:10.3390/nu15245101_

Round 1

Reviewer 1 Report

Comments and Suggestions for Authors

Authors presented the paper entitled Sleep, nutrition and injury risk in adolescent athletes: a narrative review for peer review. It is very interesting and useful manuscript not only for athletes but also for professionals dedicated to sport medicine. Adolescent athletes prone to injuries because of maturation and body rebuilding.

Several thoughts to improve the manuscript.

1. Please remove all the references from the abstract. 

2. You need to dichotomize train periods from preseason periods . Add several phrases to discuss nutrition and sleep status differences,

3. Genetic traits linking nutrition and sleep are not mentioned at all. I recommend to read works from groups guided by Garaulet Short sleep duration is associated with increased obesity markers in European adolescents: effect of physical activity and dietary habits. The HELENA study , Susan Redline https://doi.org/10.1093/sleep/33.9.1201Please add phrases about such important thing. 

Reviewer 2 Report

Comments and Suggestions for Authors

At first glance, this seems like well written manuscript. However, there are some issues that preclude this work to be published in its current form. The discussion is way too long and seems to repeat what others have already reported. The body of the text should be reduced, and some part should be re-written to promote clarity differ between what is important and redundant on the other side.

Minor issues include a) references should not be added in the abstract section, b) please add line numbers since its quite hard to comment on the text in its current form.

Reviewer 3 Report

Comments and Suggestions for Authors

The objective of this paper was to do a narrative review about sleep, nutrition and injury risk in adolescent athletes. Overall, the work concept looks very well and it is a well-documented article. I find this topic to be relevant and I believe that this is indeed an intriguing study. However, I will recommend minor revisions to this paper before publication. I feel that some further refinements would help to strengthen your paper. I list them below.

Title and Abstract

Do not include references in the abstract.

In the abstract include the objectives of the manuscript and what this manuscript add to the subject compared with other articles. 

Key words: Remove “Adolescent athletes” keyword and replace for a more suitable one.

Main part of the manuscript

I appreciate that authors include figures showing the training load framework.

When you speak about the relationship between sleep and recovery, could you put parenthesis after the numbers, as you do later in the manuscript?

It was surprising to see that you do not cite the works of Rodríguez-Negro et al when you described the differences between Adolescent and Adult Athletes. Or even the one or Romaratezabala et al. when you focus on recovery, adaptation and fatigue during the training process. http://dx.doi.org/10.3390/ijerph17165649

The manuscript provides a wide-ranging study background. Authors provided many citations in the work, which is good. However, sometimes the manuscript lacks some clarity. Could you review it to make it easier for the readers?

What does it add to the subject area compared with other published 
material?

Conclusions

The conclusion section is well organized but I would expect to see more elaborations of the ideas. 

A Limitations section would serve to address this issue.

References

References are appropriate. 

Round 2

Reviewer 1 Report

Comments and Suggestions for Authors

Authors have addressed all my issues. Thank you.